# PEBP Signaling Network in Tubers and Tuberous Root Crops

**DOI:** 10.3390/plants12020264

**Published:** 2023-01-06

**Authors:** Hendry Susila, Yekti Asih Purwestri

**Affiliations:** 1Department of Life Sciences, Korea University, Seoul 02841, Republic of Korea; 2Research Center for Biotechnology, Universitas Gadjah Mada, Yogyakarta 55281, Indonesia; 3Department of Tropical Biology, Faculty of Biology, Universitas Gadjah Mada, Yogyakarta 55281, Indonesia

**Keywords:** PEBP, tubers, tuberous roots, flowering time, underground storage organ, signaling network

## Abstract

Tubers and tuberous root crops are essential carbohydrate sources and staple foods for humans, second only to cereals. The developmental phase transition, including floral initiation and underground storage organ formation, is controlled by complex signaling processes involving the integration of environmental and endogenous cues. FLOWERING LOCUS T (FT) and TERMINAL FLOWER 1/CENTRORADIALIS (TFL1/CEN), members of the phosphatidylethanolamine-binding protein (PEBP) gene family, play a central role in this developmental phase transition process. FT and FT-like proteins have a function to promote developmental phase transition, while TFL1/CEN act oppositely. The balance between FT and TFL1/CEN is critical to ensure a successful plant life cycle. Here, we present a summarized review of the role and signaling network of PEBP in floral initiation and underground storage organ formation, specifically in tubers and tuberous root crops. Lastly, we point out several questions that need to be answered in order to have a more complete understanding of the PEBP signaling network, which is crucial for the agronomical improvement of tubers and tuberous crops.

## 1. Introduction

Billions of people around the world consume tubers and tuberous root crops. Tuber crops, such as potato (*Solanum tuberosum*) and yam (*Dioscorea* spp.), and tuberous roots or starchy root crops, such as cassava (*Manihot esculenta*) and sweet potato (*Ipomea batatas*), are important carbohydrate sources and staple foods for humans, second only to cereals [1]. Tubers and tuberous crops give more energy per unit per day than cereals while also having a high amount of dietary fibers and serving as a good source of vitamins and minerals [1]. The global production of potato, yam, cassava, and sweet potato in 2020 was 359, 74, 302, and 89 megatons, respectively [2]. Potato is mainly cultivated in subtropical and temperate regions, such as China, India, Russia, Ukraine, and the USA. In contrast, yam, cassava, and sweet potato are mainly produced in tropical regions such as Southeast Asia, Central Africa, and Middle and South America [2]. Given their widespread cultivation and high nutritional values, tubers and tuberous root crops are valued to combat food insecurity, especially in developing countries [3,4]. 

Potato and yam tubers are initiated from specialized shoots called stolons. There are two main phases of tuber formation. The first phase is stolon initiation and growth, followed by the second phase of tuber induction and growth [5]. Potato tubers mostly grow from a stolon tip that forms scattered meristematic cells from a former vasculature tissue [6]. Meanwhile, tuberous roots mainly grow from the development of the primary cambium, followed by active cell division and cell expansion in the vascular cylinder [6]. The development of underground storage organ formation is initiated by a complex molecular biology process, which is controlled by environmental factors such as nutrient content, photoperiod, and temperature [7,8,9]. 

Underground storage organs such as tubers and tuberous roots contain high carbohydrate levels produced by photosynthesis. Tubers and tuberous roots are primary sink tissues to store excess assimilates. Sucrose, as a significant photo-assimilate, is transported to the underground storage organ by SUCROSE TRANSPORTERs (SUTs), such as the high-affinity sucrose/H^+^-symporter, StSUT1, and the low-affinity sucrose/H^+^-symporters, StSUT2 and StSUT4 [10,11,12]. Furthermore, the external application of sugar induces the formation of potato tubers and cassava tuberous roots [13,14], suggesting that the initiation of underground storage organ formation is well connected with plant growth and development. 

The floral initiation and underground storage organ development are controlled by overlapping regulatory processes involving the *Phosphatidylethanolamine-Binding Protein* (*PEBP*) gene family [15]. In this review, we briefly summarize the function of PEBPs in floral initiation and underground storage organ formation of tuber and tuberous root crops, such as potato, yam, cassava, and sweet potato. We also describe the environmental and internal cues that modulate the function of PEBPs as regulators of flowering time and underground storage organ formation. Furthermore, the known upstream regulators and interacting partners of PEBPs in tubers and tuberous root crops are described and summarized into a PEBP signaling network. 

## 2. PEBPs in Tuber and Tuberous Root Crops

PEBPs were first purified from the bovine brain and characterized as having affinities with several organic anions and phospholipids, e.g., phosphatidylethanolamine (PE) [16]. The overall protein structure of plant PEBPs shows a high degree of similarity with bovine PEBP. It comprises a large central antiparallel β-sheet flanked by a smaller β-sheet on one side and an α-helix on the other side [17,18]. These two central β-sheets are conserved and essential for the anion-binding pocket structure, the signature of PEBPs [17,18].

On the basis of the genome sequence, PEBPs can be classified into three main groups, *MOTHER OF FT AND TFL1* (*MFT*)-like genes, *FLOWERING LOCUS T* (*FT*)-like genes, and *TERMINAL FLOWER 1* (*TFL1*)/*CENTRORADIALIS* (*CEN*)-like genes [19]. Algae and nonvascular plants only contain *MFT-*like genes, whereas further duplication and divergence of *FT/TFL1*-like genes have occurred in vascular plants [19]. In angiosperms, the PEBP gene family plays an important role in plant development, including floral initiation and underground storage organ formation [20,21]. In general, the florigen FT-like protein acts as an activator to promote flowering and underground storage organ formation, while the anti-florigen, TFL1/CEN-like protein, has a function to repress flowering and underground storage organ formation [15,22]. In Arabidopsis, *FT, TWIS SISTER OF FT* (*TSF*)*,* and *MFT* are known as floral activators, while *BROTHER OF FT AND TFL1* (*BFT*)*, TFL1,* and *ARABIDOPSIS THALIANA CENTRORADIALIS* (*ATC*) act as floral repressors [23]. Interestingly, the functional evolution of FT-like genes likely started in basal angiosperms, as the gymnosperm FT-like genes failed to promote flowering, in contrast with the gymnosperm TFL1-like genes, which were capable of repressing flowering when expressed in *Arabidopsis* (*Arabidopsis thaliana*) [24,25].

Several copies of *FT-*, *TFL1/CEN-*, *BFT-*, and *MFT-*like genes were identified in the selected tubers and tuberous root crops, including potato, yam, cassava, and sweet potato (Figure 1A). On the basis of phylogenetic analysis, we found that potato has three *FT-*like genes, two *TFL1/CEN-*like genes, three *BFT*-like genes, and one *MFT-*like gene, consistent with a previous report [26]. We found that yam (*Dioschorea rotundata*) has eight *FT-*like genes, three *TFL1/CEN*-like genes, one *BFT-*like gene, and one *MFT-*like gene. Cassava has two *FT-*like genes, five *TFL1/CEN-*like genes, one *BFT-*like gene, and two *MFT-*like genes. Lastly, sweet potato has 10 *FT-*like genes, 11 *TFL1/CEN-*like genes, and seven *MFT-*like genes. Among the identified *FT-*like genes, until now, only *PGSC0003DMG400016179* (*StSP3D*) and *Manes_12G001600* (*MeFT1*) have been characterized as floral activators [21,27,28], while *PGSC0003DMG400023365* (*StSP6A*) was found to be responsible for promoting potato tuberization [21]. Interestingly, *StSP5G*, the repressor of *StSP6A* [29], is seemingly not annotated by the Potato Genome Sequencing Consortium (PGSC). Furthermore, our phylogenetic analysis suggested that *PGSC0003DMG400014322* (*StCEN*), which has been identified as a negative regulator of floral development and tuber formation [30,31], is more closely related to *BFT* than *TFL1/CEN* (Figure 1A). 

The divergence of FT-like genes involves a positive selection of the fourth exon after the duplication event [33]. Among the thirty-nine nonconservative substitutions that distinguish FT and TFL1, four residues (E109, W138, Q140, and N152) in the divergent external loop segment in the fourth exon are critical for the opposite function of FT and TFL1 [18,34]. Several FT homologs undergo neofunctionalization after the duplication event to revert the function of FT from floral activator into floral repressor [35]. For example, an FT homolog from sugar beet (*Beta vulgaris*), BvFT1, acts as a floral repressor due to mutations that correspond to Y134N, G137Q, and W138Q in *Arabidopsis* FT [36], while barley (*Hordeum vulgare*), HvFT4, acts as a floral repressor likely due to W138A mutation [37]. Other residues such as V70, S76, and R83 are crucial for cell-to-cell movement of FT near the shoot apical meristem (SAM) region [38], while the D73, E84, Y85, P94, R119, and G171 residues are essential for the function of FT as the coactivator [20,34].

The majority of FT-like proteins from tubers and tuberous root crops have conserved critical residues in comparison with *Arabidopsis* FT (Figure 1B). Interestingly, PGSC0003DMG400016180 (*StSP5G-*like) has two mutations corresponding to R83K and W138N in *Arabidopsis* FT which are crucial for FT movement [38] and neofunctionalization of FT as a floral repressor [34], respectively. A similar mutation, crucial for the neofunctionalization of FT as a floral repressor, is also seen in the yam. For instance, compared with *Arabidopsis* FT, DRNTG_29881 has a Y85H mutation, DRNTG_11086 has a W138M mutation, DRNTG_11087 has W138M and Q140H mutations, and DRNTG_20845 has a Q140P mutation. Furthermore, DRNTG_16474 has a V70I mutation, which is crucial for FT movement [38]. Therefore, only two FT-like proteins, DRNTG_28722 and DRNTG_10592, are likely responsible for promoting flowering and underground storage organ formation in yam. For tuberous root crops, two FT-like proteins in cassava and 10 FT-like proteins in sweet potato have similar critical residues to *Arabidopsis* FT. Nevertheless, functional studies of the remaining *FT-, BFT-*, and *TFL1/CEN-*like genes in potato, yam, cassava, and sweet potato are essential to understand the genetic redundancy and the signaling network of PEBPs in the regulation of flowering time and underground storage organ formation.

## 3. PEBP Signaling Network in Flowering Time Regulation

Internal and external cues strictly control the timing of flowering by regulating *PEBP* [39] (Figure 2A). Long-day (LD) plants, such as *Arabidopsis*, flower in response to increasing day length, whereas other plants, including rice and sorghum, flower under short-day (SD) conditions. Photoperiodic flowering is controlled by the integration of light perception and internal circadian rhythm [40]. In *Arabidopsis*, LD conditions induce high levels of *FT* and *TSF* expression, which eventually promote flowering [20]. Transcription factor complexes are also involved, such as the B-box transcription factor CONSTANS (CO) [41,42]. The rhythmic expression and protein stability of CO are controlled by the circadian clock and light, which eventually only allow *FT* and *TSF* expression at dusk under LD conditions [39,40]. In contrast, the expression of *ATC* is activated by SD conditions, especially at night time, likely to prevent flowering under noninductive SD conditions [43].

Temperatures also control the flowering time of Arabidopsis by flowering earlier under warm temperatures and later under cool temperatures [44] (Figure 2A). Warm temperatures promote the expression of *FT* and *TSF* through the activation of *PHYTOCHROME INTERACTING FACTOR 4* (*PIF4*) [45]. Moreover, temperature regulates FT trafficking through the modulation of FT–phospholipid interactions [46]. Drought also accelerates flowering through the drought escape response by activating *FT* and *TSF* expression via phytohormone abscisic acid (ABA) [47,48]. Interestingly, early drought also activates *BFT* expression, which likely serves as a buffering mechanism to counteract *FT* and *TSF* activation [49]. The phytohormone gibberellic acid (GA) also promotes flowering through *FT-*dependent and -independent pathways [50]. Lastly, other internal cues, including sugars as photosynthesis products and age, regulate flowering through microRNA 156 and microRNA 172 via *FT-*dependent and -independent pathways [51] (Figure 2A). 

*FT* is expressed in specific cells in the leaf vasculature called companion cells, in contrast with *TFL1*, which is expressed in the SAM [52,53] (Figure 2A). FT is then translocated to the SAM through interactions with several multiple C2 domain and transmembrane region proteins (MCTPs) transporters, including FT-INTERACTING PROTEIN1 (FTIP1/MCTP1), QUIRKY (QKY/MCTP15), and MCTP6 [54,55,56]. In the SAM, both FT and TFL1 compete to interact with chromatin-bound transcription factors FD to modulate the expression of floral identity genes, such as *SUPPRESSOR OF OVEREXPRESSION OF CO 1* (*SOC1*), LEAFY (*LFY*), *FRUITFULL* (*FUL*), and *APETALA1* (*AP1*) [57]. Furthermore, *Arabidopsis* BFT, TSF, and ATC were also reported to interact with FD [43,58,59], while the MFT ortholog from *Adiantum capillus-veneris* is also capable of interacting with FD [60]. The interaction of FT or TFL1 with FD is mediated by the 14-3-3 protein to form either a floral activator complex or a floral repressor complex [61]. Considering that all *Arabidopsis* PEBPs contain the 14-3-3-binding motif, this suggests a dynamic complex formation to fine-tune floral formation (Figure 2B). Furthermore, FT and TFL1 also interact with different classes of TCP (for TEOSINTE BRANCHED1, CYCLOIDEA, and PCF) transcription factors to regulate flowering time [34,62]. 

Unlike in the model plant *Arabidopsis*, the molecular mechanism of flowering time regulation in tubers and tuberous root crops is still elusive [22]. Most tubers and tuberous root crops are mainly propagated through a vegetative method using clonal propagation from a tuber, stem, or vine [63,64,65,66]. Clonal propagation through the tuber or stem is the easiest way to multiply the plants as it does not require a complicated technique while maintaining the parental plants’ characteristics. However, loss of genetic diversity, deleterious mutation, and pathogen accumulation are the major pitfalls of these clonal propagation methods [67]. Furthermore, the relatively long growth period of tubers and tuberous crops is a major obstacle in the breeding process of these plants [68,69,70]. Therefore, the complete understanding of molecular mechanism that control floral formation in tubers and tuberous root crops could aid in the breeding process of these crops.

Potato can be classified into two subspecies, the wild Andean varieties (*S. tuberosum* ssp. *andigena*), cultivated in the Andean highlands, and the modern potato (*S. tuberosum* ssp. *tuberosum*), originating from the lowlands of southern Chile [71]. The flowering time of both potato varieties is accelerated under LD conditions, although the plant still flowers under short day (SD) conditions [72,73]. Interestingly, unlike in *Arabidopsis*, *StSP3D* is expressed in the middle of the day instead of at dusk [74]. Furthermore, constitutive expression of *Arabidopsis CO* results in downregulation of *StSP3D* and late flowering in potato plants [21,75]. Potato *StCO* is encoded by three tandem genes, *CONSTANS-like 1* (*StCOL1*), *StCOL2,* and *StCOL3* [29]. The knockdown of *StCOL1* using RNA interference accelerates potato flowering time [29], in contrast with the role of CO as a floral activator in *Arabidopsis* [42]. StCOL1 directly activates the expression of *StSP5G* through the TGTGGT motif in the *StSP5G* promoter [29]. StSP5G represses the expression of *StSP3D* and, thus, delays flowering [21,29] (Figure 3A).

The potato *B-box 24* (*stBBX24*) acts as a negative regulator of the flowering time through the regulation of *StSP3D* (Figure 3A), in contrast with the positive role of *BBX24* in *Arabidopsis* flowering [74,76]. However, the overexpression of potato *CYCLING DOF FACTOR 1* (*stCDF1*) results in a late flowering phenotype [77], consistent with the role of *CDF1* as a floral repressor in *Arabidopsis* [78,79]. Interestingly, while *Arabidopsis* CDF1 prevents flowering through repressing *CO* and *FT* transcription [80,81], StCDF1 prevents flowering by activating the tuberigen, *StSP6A* [77] (Figure 3A). *StSP6A* was found to prevent potato flowering, as *StSP6A* knockdown plants flowered early, while plants with high *StSP6A* levels flowered late [77]. The flowering time regulation by *StSP6A* seems unrelated to the regulation of *StSP3D* expression [21,77]. It is noteworthy to mention that StSP6A still maintains florigenic activity, as the overexpression of *StSP6A,* using constitutive *35S* promoter, accelerated the flowering of *Arabidopsis* and potato [21]. Considering that both *StSP6A* and *StSP3D* are expressed in the leaves and stem, it is interesting to speculate whether StSP6A competes with StSP3D to interact with protein transporters such as the MCTPs [54] (Figure 3A). 

LD conditions also promote the flowering of yam and cassava, while sweet potato flowering is accelerated under SD conditions [82,83,84,85]. The upregulation of *MeFT1* and *MeFT2* likely regulates the photoperiodic flowering of cassava under LD conditions [86] (Figure 3B). However, the function of MeFT2 to promote flowering still needs to be discovered. Furthermore, we have yet to find any report describing the role of florigen or anti-florigen, as well as any molecular study of flowering time regulation in yam and sweet potato. Considering that the regulation of flowering time by photoperiod is likely caused by natural selection and domestication of crops [15,87], further research into this area may uncover novel insight into photoperiodic flowering in tubers and tuberous root crops.

In addition to the photoperiod, temperature is one of the seasonal cues that control the flowering time of many plant species [39]. A cold night temperature (5–6 °C) or GA treatment during SD conditions is reported to induce the flowering of LD potato species, including *S. sparsipilum, S. acaule, S. punae,* and *S. demissum* [72]. In *Arabidopsis,* the expression levels of the GA biosynthesis gene, *AtGA20ox1, AtGA20ox2,* and *AtGA30x1* are induced under low temperatures [88]. Cold night temperature also reduces the expression of the GA catabolism gene, *SlGA2ox*, and promotes the accumulation of bioactive GA in tomato (*S. lycopersicum*) [89]. Furthermore, the flowering time of potato is also accelerated by high light, albeit in an *StSP3D-*independent manner [90]. In *Arabidopsis*, high light prevents the transcription of the floral repressor, *FLOWERING LOCUS C* (*FLC*), through nuclear accumulation of the PHD type transcription factor [91,92]. Considering that FLC also regulates temperature-responsive flowering of *Arabidopsis* and GA biosynthesis and signaling [93,94], it is tempting to speculate that temperature, high light, and GA also mutually regulate the flowering time of potato. 

Cassava’s flowering time is induced by LD conditions and cool temperature (early flowering at 22 °C and late flowering at 34 °C) [84,95] (Figure 3B). The temperature-responsive flowering of cassava has the opposite pattern to several plant species, including *Arabidopsis*, which flower early in warm temperatures [44]. However, unlike in *Arabidopsis,* the expression levels of *MeFT1* and *MeFT2* are mostly unperturbed by temperature, while *PIF4* expression increases in warm temperatures (similar to *Arabidopsis*) [84,86]. As shown in another study, the flowering time of field-grown cassava in Southeast Asia is promoted during the dry season in mountainous regions [96]. The drought and low night temperature of mountainous regions activate the expression of *MeFT1* and *MeFT2* [96] (Figure 3B). The activation of florigen by drought is well documented in annual plants, including *Arabidopsis* and rice [47,48,97]. 

Overall, the mechanism underlying the long-distance transport and activation of the downstream targets of florigen in tubers and tuberous root crops is still elusive. However, a recent report suggested that StSP3D is able to form a floral activation complex with StABI5-like 1 (StABL1) to promote flowering [98]. However, whether StSP3D also makes a complex with StFDs such as StSP6A needs further verification. To initiate floral formation, the StSP3D/St14-3-3/StABL1 complex directly activates the expression of *StSOC1* and *StFUL* to mediate meristem determinacy [98]. Furthermore, the anti-florigen StCEN is also capable of making a complex with StFD-like (StFDL1), probably to repress floral formation [31] (Figure 3A). Considering the sequence conservation of PEBPs, it is tempting to speculate that florigens and anti-florigens in other tubers and tuberous root crops may also make complexes with 14-3-3, ABI5-like, and FD-like proteins. Nevertheless, further analysis of upstream regulators of PEBPs, their transporters, and their downstream targets is essential to expand the PEBP signaling network in tubers and tuberous root crops.

## 4. PEBP Signaling Network in Underground Storage Organ Formation

Underground storage organ formation, including tubers and tuberous roots, is essential for plant survival under adverse environmental conditions. Day length regulates potato tuber formation in a genotype-dependent manner. For instance, the wild Andean varieties strictly require SD conditions to promote tuber formation, while the modern potato cultivar is more adapted to LD conditions [71]. SD conditions also promote the tuber formation of yams (*D. rotundata, D. alata,* and *D. cayanensis*), while LD conditions inhibit tuber formation and stimulate vine and leaf growth [99,100]. Interestingly, SD conditions also promote the tuberous root formation of cassava [101], suggesting an inverse correlation of photoperiod requirement for floral initiation and underground storage organ formation.

Similar to flowering time, photoperiodic tuber formation is controlled by integrating light signaling and the circadian clock [71]. Light signals are perceived by multiple photoreceptors, including phytochromes, cryptochromes, phototropins, and F-box-containing flavin-binding proteins [102]. White- or red-light treatment of SD-grown plants in the middle of the night (known as night break) represses the tuber formation of Andigena potatos, while subsequent application of far-red light reverses the repression [103], suggesting the role of phytochrome in tuberization. Indeed, the knockdown of *StPhyB* induces tuber formation in noninductive conditions (LD and SD + night break) [104]. Furthermore, StPhyB interacts with StPhyF to repress tuber formation and flowering by stabilizing StCOL1 [29,105] (Figure 3A), in contrast to their function in *Arabidopsis*, where PhyB induces CO degradation [106]. Interestingly, *Arabidopsis* CONSTANS-like 7 is stabilized by PhyB [107], similar to StCOL1, suggesting that the function of PhyB depends on the CO-like protein sequence. StCOL1 then represses tuber formation by activating *StSP5G* transcription, a negative regulator of *StSP6A* [29]. Finally, StSP6A undergoes long-distance movement to initiate tuber formation [21]. Interestingly, the photoperiodic tuberization in water yam (*D. alata*) is also likely connected with *FT-like* genes, *DaFT1* and *DaFT2* [108].

The natural variation of *StCDF1* confers the adaptation of European potato to tuberization under noninductive LD conditions [109]. StCDF1 represses the expression of *StCOL1* and *StCOL2* to activate tuber formation [29,109]. The early-maturing line, CE3130, has two truncated alleles, *StCDF1.2* and *StCDF1.3*, which do not contain a carboxyl-terminal region that is important for interaction with FLAVIN-BINDING, KELCH REPEAT, F-BOX1 (FKF1) [109]. StFKF1 and GIGANTEA (StGI) promote the degradation of StCDF1 through direct interactions [109] (Figure 3A), suggesting the similarity of molecular components of the photoperiodic response between potato and *Arabidopsis.* Furthermore, the BELLRINGER-1-like transcription factor (StBEL5) and its interacting partner, the StKNOX transcription factor, have been identified as activators of *StCDF1* and *StSP6A* [110,111]. 

The expression levels of *StBEL5* are regulated by photoperiod through the PhyB/microRNA miR172 cascade [112]. Similar to flowering time regulation in *Arabidopsis*, miR172 and miR156 have been shown to promote and inhibit tuber formation, respectively, partly through the regulation of *StSP6A* [112,113]. Both miR172 and miR156 are graft-transmissible and regulate phytohormone signaling, including GA, cytokinin, and strigolactone [112,113,114]. Considering that the sequential action of miR156 and miR172 define the age-dependent pathway of flowering time regulation in *Arabidopsis* [51] (Figure 2A), it is tempting to speculate whether a similar mode of action also regulates underground storage organ formation. 

Temperature is another seasonal cue that controls underground storage organ formation. Warm temperatures inhibit potato tuber formation and sweet potato and Mexican turnip (*Pachyrhizus tuberosus*) tuberous root formation [112,115,116]. Warm temperature promotes the expression of a small RNA called *Suppressing Expression of SP6A* (*SES*), an upstream regulator of *StSP6A* [117]. *SES* reduces the accumulation of *StSP6A* transcripts to reduce the sink strength and delay tuber formation [117]. Interestingly, *SES* is likely a potato-specific small RNA, as no homolog of *SES^-pri-miRNA^* was found in the Solanaceae genome [117]. Furthermore, a circadian clock component, TIMING OF CAB EXPRESSION 1 (StTOC1), interacts with StSP6A to inhibit the autoactivation of *StSP6A*, especially in warm temperatures [118]. In *Arabidopsis*, the interaction between TOC1 and PIF4 mediates the circadian gating of thermomorphogenesis [119], suggesting that a similar mechanism may apply to underground storage organ formation. Further analysis of temperature-sensing components, including PhyB, *EARLY FLOWERING 3*, and florigen movement, could potentially elucidate the molecular mechanism of temperature-responsive underground storage organ formation [44,46,120]. 

A previous report suggested that the tuberigen signal, StSP6A, prevents floral formation in an StSP3D-independent manner [77]. However, the relationship between floral initiation and underground storage organ formation remains elusive. Interestingly, the removal of flowers increases the yield of potato and artichoke (*Helianthus tuberosus*) tubers [121,122,123], suggesting a competition between flower and tuber development for photosynthesis products. Furthermore, the overexpression of *FT* reduces the tuberous root growth and accelerated flowering of cassava, suggesting a relationship between both developmental processes [124] (Figure 3B). On a molecular level, the source–sink regulation of tuber formation is controlled by the interaction between StSP6A and sucrose efflux transporter StSWEET11 in the plasma membrane [125]. The binding of StSP6A to StSWEET11 prevents the leakage of sucrose to the apoplast, promoting symplastic sucrose transport and activating tuber formation [125]. The presence of StSP6A along the phloem also likely enables enhanced sucrose delivery toward tuber development [126] (Figure 3A). It will be interesting to see whether StSP3D also interacts with StSWEETs to compete in source–sink carbon partitioning for flower development. Furthermore, whether FT-like protein is also involved in sucrose export during tuberous root formation will be interesting to study (Figure 3B).

Similar to floral initiation, StSP6A forms a tuberigen activation complex by making a protein complex with St14-3-3 and StFD-like1 (StFDL1) to activate tuber formation [27]. Moreover, StCEN is also reported to interact with StFD and StFDL1a to prevent the activation of StFD and StFDL1 downstream targets, including *StMADSs*, *StSP6A*, and *germin3* [31]. StSP6A and StCEN likely compete to form tuberigen activation or repression complex to control tuber formation [31] (Figure 3A), similar to the competition between FT and TFL1 during *Arabidopsis* floral formation [57] (Figure 2B).

The phylogenomic approach suggests that StSP6A and its interacting partner, IDENTITY OF TUBER 1 (IT1), a plant-specific TCP transcription factor, are crucial for tuber formation in potato species [127]. Another TCP transcription factor, BRANCHED1b, interacts with StSP6A to prevent aerial tuber formation [128] (Figure 3A). In *Arabidopsis*, BRANCHED1 interacts with FT but not TFL1 to prevent floral transition in the axillary meristem [129], suggesting a conserved regulatory network across plant species. StSP6A was also found to interact with Flowering-Promoting Factor 1, No Flowering in Short Days 1 (StNFL1), and StNFL2, which are likely crucial for tuber formation [130]. 

StSP6A also forms an alternative complex with St14-3-3 and StABL1 to promote tuber formation [98]. StABL1 binding regions are enriched with several binding motifs, including the bZIP/BHLH and TCP transcription factor-binding motifs, suggesting that the StSP6A/St14-3-3/StABL1 complex may form a larger protein complex with TCP transcription factors, such as IT1 [98,127]. StABL1 also directly activates the expression of *StGA2ox1*, a GA-catabolizing enzyme [98]. The fact that StABL1 is a component of ABA signaling probably explains the opposite role of GA and ABA as an inhibitor and an inducer of tuberization, respectively [98,131]. Interestingly, cassava tuberous root formation is inhibited by ABA [132], suggesting the opposite role of phytohormone signaling in (potato) tuber and (cassava) tuberous root development. Further functional studies of yam, cassava, and sweet potato PEBP are essential to expand our knowledge on the molecular mechanism of underground storage organ formation of these plants.

## 5. Conclusions and Future Perspectives

Floral initiation and underground storage organ formation are complex processes that integrate environmental cues and endogenous signals through the regulation of master regulators, i.e., the *PEBP* gene family [6,22]. FT and TFL1 act oppositely due to the external loop segment variation in the fourth exon [18]. Subtle amino-acid changes in the external loop segment during evolution and diversification cause neofunctionalization of PEBP, converting FT from a coactivator into a corepressor [15,35,36]. The PEBP signaling network of tuber and tuberous root crops is likely conserved, as upstream regulators and interacting partners of FT- and TFL1-like proteins are likely similar such as in *Arabidopsis* [6,15,22,39,71]. Several important issues, as listed below, could offer a future perspective to elucidate the PEBP signaling network in tubers and tuberous root crops.
All *Arabidopsis* PEBPs are involved in flowering time regulation [15,23], while little is known in other plant species. Furthermore, whether PEBPs also regulate tuberous root formation remains elusive. Therefore, a functional study of the role of *PEBP* in flowering time and underground storage organ formation is essential to elucidate the PEBP signaling network in each tuber and tuberous root crop;Further in-depth studies of the molecular mechanism that controls the function of PEBP are needed. For example, the non-cell-autonomous function of FT requires an interaction with the MCTP transporter and phospholipids [24,46,55,56,133]. However, the mechanism underlying the long-distance transport of FT-like proteins in tubers and tuberous root crops is still elusive. Therefore, further verification of protein transporters and phospholipid interactors of the PEBP family from tubers and tuberous root crops is needed for a complete understanding of the PEBP signaling network in these plants. Furthermore, although StSP5G is a central regulator connecting the photoperiod with floral initiation and tuber formation by regulating *StSP3D* and *StSP6A* [21,29,109], it is not clear how StSP5G regulates its downstream targets. StSP5G may interact with other proteins, including St14-3-3s and StTCPs, to modulate floral initiation and tuber formation. Therefore, translating the knowledge from *Arabidopsis* could accelerate the characterization of the regulatory module of the *PEBP-*gene family in tubers and tuberous root crops (Figure 2);Some plants develop underground storage organs to survive adverse environmental conditions [6]. Whole-genome analysis of the non-tuber-bearing *Etuberosum*, sister of the *Petota* (tuber-bearing) section of the *Solanum* genus, suggests that the non-tuber-bearing phenotype of *Etuberosum* is caused by the deletion of the fourth exon of *SP6A* [127]. However, a more in-depth examination is needed to elucidate how PEBP, a flowering time regulator, evolved and diversified its function to regulate underground storage organ formation.

## Figures and Tables

**Figure 1 plants-12-00264-f001:**
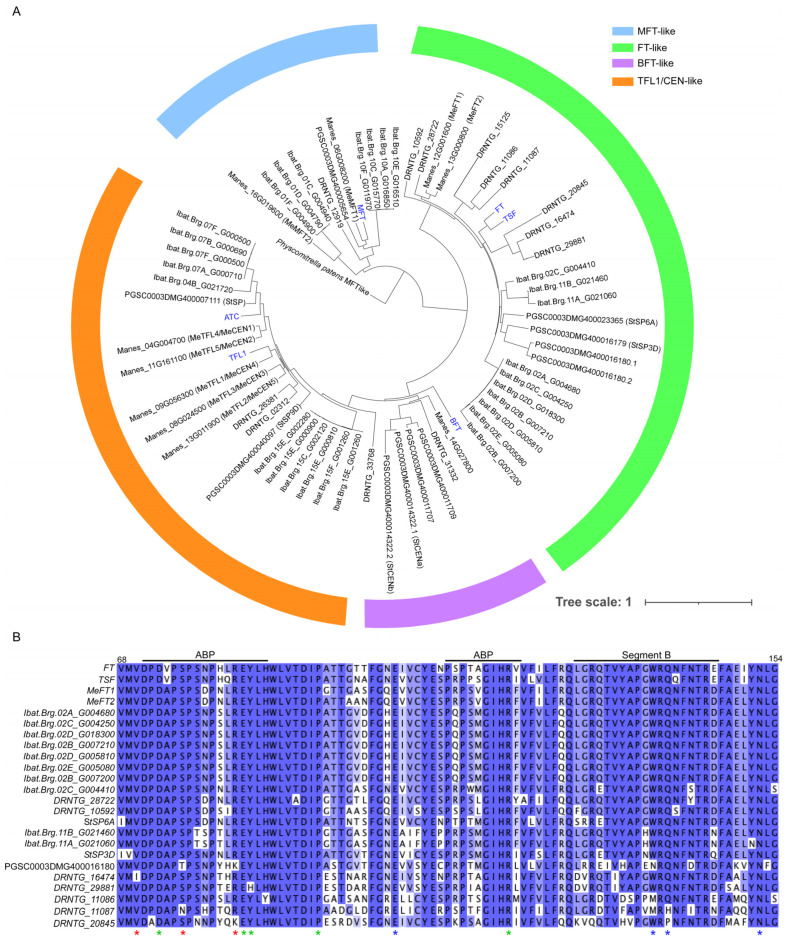
Characterization of PEBP family in tubers and tuberous root crops. (**A**) Phylogenetic tree of the PEBP family in potato, cassava, and yam. The tree was constructed using webtool Phylogeny.fr with the maximum likelihood method [32]. The protein name represents the Ensembl Plants accession number. (**B**) Multiple sequence alignment of the deduced amino-acid sequences of FT for the conserved anion-binding pocket (ABP) and segment B of the fourth exon. Red, green, and blue asterisks represent amino-acid residues that are important for FT movement, FT function, and FT-to-TFL1 neofunctionalization, respectively.

**Figure 2 plants-12-00264-f002:**
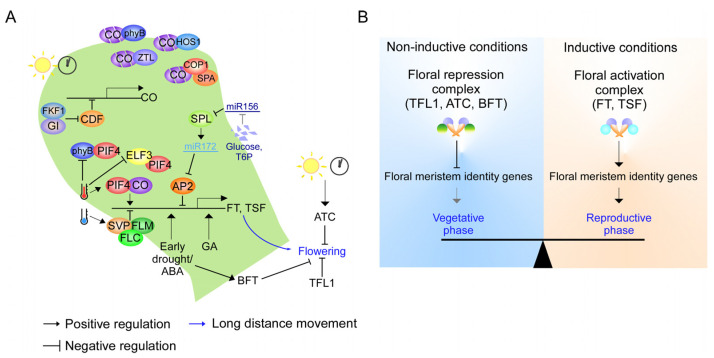
Schematic representation of PEBP signaling network in flowering time regulation of *Arabidopsis.* (**A**) Schematic representation of flowering time regulation in *Arabidopsis* by PEBP. Internal and external cues regulate the expression and activity of PEBP to fine-tune floral development. LD conditions, warm temperatures, early drought, ABA, GA, and sugars represent positive signals to promote floral initiation. In contrast, SD conditions and cool temperatures represent negative seasonal cues that delay floral formation. (**B**) Classification of *Arabidopsis* PEBP as a floral activator or floral repressor. Under noninductive conditions, FD and 14-3-3 primarily form a floral repression complex with anti-florigen, TFL1, ATC, and BFT to repress the expression of floral meristem identity genes. However, under inductive conditions, high levels of florigen, FT, and TSF compete with anti-florigen to make a floral activation complex that activates floral meristem identity genes.

**Figure 3 plants-12-00264-f003:**
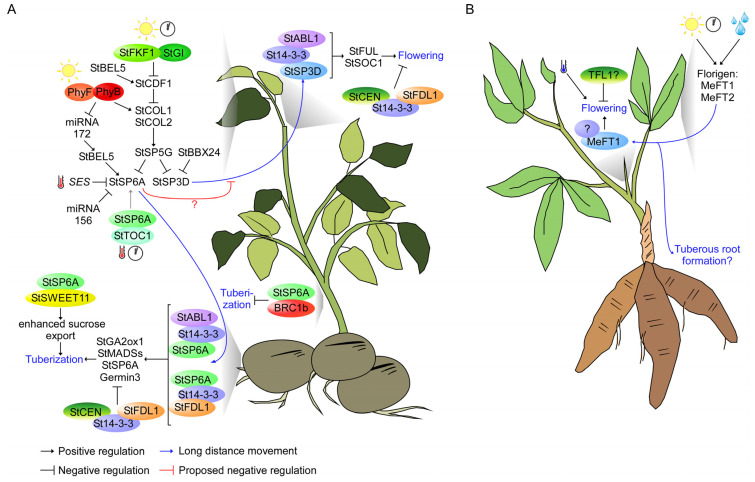
Schematic representation of PEBP signaling network in potato and cassava. (**A**) Schematic representation of flowering time and underground storage organ formation in potato. Internal and external cues modulate PEBP expression, including *StSP5G, StSP3D,* and *StSP6A*. StSP3D undergoes long-distance movement to the SAM to initiate flowering, while StSP6A moves to the stolon to promote tuber formation. Another PEBP, StCEN, acts oppositely to prevent floral initiation and tuber formation. (**B**) Schematic representation of flowering time and underground storage organ formation in cassava. External cues, including photoperiod and drought, promote the expression of florigens, *MeFT1* and *MeFT2*, to modulate flowering time.

## Data Availability

This review does not report novel data.

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
