# Peer review of "PEBP Signaling Network in Tubers and Tuberous Root Crops"

_plants, 2023, doi:10.3390/plants12020264_

Round 1

Reviewer 1 Report

Dear authors,

The Phosphatidyl Ethanolamine Binding Protein (PEBP) gene family is present in all eukaryote kingdoms, which are involved in various biological processes. In this review, the authors reviewed the molecular signaling network of PEBP in floral initiation and underground storge organ formation. These results should help us to clarify the physiological and molecular mechanisms controlling the growth of tuber and tuberous crops, and to improve the quality and yield. However, the manuscript still has some minor problems that need further revision to improve its quality.

1.     The title of the article is PEBP signaling network in the tuber and tuberous root crops. From the section of PEBP signaling network in flowering time regulation (Line 137 to 181), most of the content statements have nothing to do with tuber and tuberous root crops, only the molecular mechanism of flowering regulation of PEBP gene family in other crops such as Arabidopsis thaliana. There are many similar problems later in the following text. Although the content is very detailed, it does not match the main idea of the manuscript. Therefore, please revise the manuscript around the main idea, don't quote too much.

2.     In the section of PEBP signaling network in underground storage organ formation, there are also some contents (from 285 to 296) that have little relevance to the topic.

Reviewer 2 Report

 The review authors have presented is extense and present a good up to date recopilation of PEBP signalling network. I have several concerns but are almost all related with formal details and the use of some expressions in the main text that I guess could authors to get a little better final version:

Authors must review coherence in the paragraphs, as in many cases several sentences do not reflect a linear coherent description. For instance, from lines 182 to 194: first they talk in general for negative effect of clonal propagation, then they go to a single example (till line 190) and then "The pathogen resistance cultivar has been developed...", as sentence is in singular and inmediately after the particular example of potato  you expect authors are talking about a cultivar developed to resist Phytophthora infestans, however the references just talk in general as possible strategies that can do it "in the future". Actually one of the refence is in cassava and the other in tomato, and the use of "has been developed" meens is already done, but papers are suggestions for the future. Moreover, with no indication of going again to tuberous crops in general, any reader will be confused.

I do not understand neither the expression in the lines 180-181. In the middle of section 3, a sentence that says what is summarized in the section... Instead of "section" maybe be authors wanted to say "image/figure/scheme". Even in this case the sentence is weird. Usually, the first time some of the information summarized in the figure is presented in section 3 is the moment to indicate "Figure 3" and anyone understand that the rest of information in the section is obviuosly related with this figure unless a new one is indicated.

Similarly in lines 297 and 298, where I do not understand the meaning of the sentence in that position.

Line 200: "StSP3D are expressed... instead at dusk" >> "StSP3D is expressed... instead of at dusk"

Line 222: For me is weird to read the expression "accelerated the flowering of Arabidopsis and tomato", is a consense to talk about flowering as the number of leaves till bolting in Arabidopsis for instance, more than absolute time as many aspects can affect absolute time and experiments in different labs would be difficult to compare. Using the verb "accelerate" give's the impression of difference in time. Please revise data in Navarro et al. and use the correct expression. I prefer the verb "reduce" than "accelerate" to express a shorter flowering time. I have read the same use of "accelerate" in other parts of the manuscript.

Line 403: "All Arabidopsis PEBP involved in..." > "All Arabidopsis PEBP are involved in..."
